# Age as a determining variable in the practice or abandonment of physical exercise among young people

**Iago Portela-Pino**[1], **Teresa Valverde-Esteve**[2]*, **María José Martínez-Patiño**[3]

**1** Department of Health Sciences, University Isabel I, Burgos, Spain, **2** Department of Didactics of Music, Visual and Body Expression, University of Valencia, Valencia, Spain, **3** Department of Specific Didactics, University of Vigo, Vigo, Spain

☙ These authors contributed equally to this work.
* Teresa.valverde@uv.es

**Data Availability Statement:** All relevant data are within the manuscript.

**Funding:** TVE: Conselleria de Inno-vación, Universidades, Ciencia y Sociedad Digital

## Abstract

### Background

Numerous studies have reported a gradual decline in the practice of physical exercise with age, confirming high dropout rates in the adolescent period. The aim of this study was to investigate the motivation or barriers that lead to greater or lesser adherence to physical exercise among children and adolescents.

### Methods

In this study, 1,081 individuals volunteered to participate [239 children (age: 10.89 ± .66 years) and 842 adolescents (age: 14.9 ± 1.75 years)]. For this purpose, we used the Self-Report on Motivation for Exercising (AMPEF) and Self-Report on Barriers to Exercising (ABPEF) questionnaires.

### Results

The most significant motivational factors in the practice of physical exercise (in both children and adolescents) are competition, social recognition and challenge ($t = -16.02$, $p < .001$), prevention and positive health ($t = 5.24$, $p < .001$) and affiliation, fun and well-being ($t = -19.54$, $p < .001$), while the barriers focus on fatigue and laziness ($t = -13.20$, $p < .001$) and body image and physical-social anxiety ($t = -5.71$, $p < .001$). It is also observed that adolescents showed significantly higher motivation than children and greater presence of barriers in the practice of physical exercise.

### Conclusions

The age is a determining variable to include in preventive and intervention programs for the practice of physical exercise.

(Generalitat Valenciana, Best/2019/110), Fulbright Comission and University of Valencia grants.

**Competing interests:** The authors have declared that no competing interests exist.

## Introduction

Healthy lifestyles, understood as a person's set of guidelines and everyday behavioral habits, are the basis for maintaining health [1, 2]. Thus, the practice of physical activity is deeply related to quality of life and satisfaction with it [3, 4], being particularly important in the field of education and health.

Consequently, in order to achieve a better quality of life and increase self-esteem, it is essential to know about changes in behavior, action planning, identification of barriers, resolution of difficulties or motivational aspects that lead children and adolescents to increase the performance of physical leisure activities [5, 6]. In fact, at this stage they are still very impressionable and influenceable and it constitutes a key period for the acquisition of habits and routines [7, 8].

Despite it being recognized at a social level that the elimination of sedentary habits significantly improves health, there are still impediments that hinder participation in physical activity programs [9]. The studies carried out by Hoare et al. [10] in adults give as the most common motivations: losing or maintaining weight, avoiding or controlling their state of health and improving their appearance. The most commonly mentioned barrier, however, is lack of time. Research in populations of children and adolescents show fun as the most decisive motivating factor and lack of time as the main barrier in the practice of physical exercise [11–13].

Although there is a decline in the levels of physical exercise for all age groups, there are numerous studies that show a gradual decline in the practice of physical exercise with age, confirming high dropout rates in the adolescent period [14–16]. Likewise, adolescence has been considered a negative period in the practice of physical exercise [17]. However, physical activity in childhood and adolescence is considered a primary factor in the reduction of diseases later in life [18]. There is scientific evidence showing that the effects of doing physical exercise during childhood or adolescence last into later life [19].

It is important to emphasize that the arguments made about the reduction of physical exercise with age do not remain clear if there is consensus as to the combination of biological, psychosocial and cultural factors being the main trigger for abandoning or reducing the practice of physical exercise [16]. Undoubtedly, the benefits of physical exercise at any age mainly focus on increasing the person's life expectancy, improving their physiological state and fundamentally enriching their quality of life [20, 21]. Likewise, the positive relationship between physical exercise and individuals' attention, concentration, expectations of success, subjective value of the task, self-concept, attitudes and behaviors has been pointed out [22, 23]. However, despite the benefits argued, indicators of physical inactivity and its consequences continue to be alarming [24].

The conviction of the positive role played by physical exercise in the prevention of numerous diseases is verified [25, 26], supporting the main purpose of this study, which is to investigate the motivations or barriers that lead to greater or lesser adherence to physical exercise among preadolescents and adolescents and to check if there are significant differences between the two periods. In addition, physical and sports activities not only meet health benefits but also social [27]. We must not forget the role of the sport in achieving the Sustainable Development Goals (SDG) as an effective Tool for promoting peace and development [28–33]. The Resolution 70/1 of the United Nations General Assembly, entitled "Transforming our world: the 2030 Agenda for Sustainable Development" and approved in 2015, also recognizes the role of sport in promoting social progress, noting its importance to promote "tolerance and Respect, and which also supports the empowerment of women and youth, people and communities, as well as the Goals in matters of health, education and social inclusion." In this sense, in

recent years, different institutions have proposed experiences to promote sport related to Objectives 3: guaranteeing a healthy life and promoting the well-being of all at all ages; 4: ensuring inclusive and equitable quality education and promoting lifelong learning opportunities for all; 5: Achieve gender equality and empower all women and girls; 11: Make cities and human settlements inclusive, safe, resilient and sustainable; 16: Promoting Just, Peaceful and Inclusive Societies and 17: Revitalizing the Global Partnership for Sustainable Development. Thus, its barriers and limitations have been described [34–36], thus suggesting analysis models based on the practice [37]. We can expect higher motivation and greater absence of barriers during adolescence than during preadolescence (hypothesis 1). Similarly, taking into account the findings of previous studies, a strong correlation between motivational factors and barriers in the practice of physical exercise is predicted in both periods (hypothesis 2).

## Methods

### Participants

This study follows a positivist methodological approach. To carry out this empiric, descriptive and comparative research [38] we worked with two non-probabilistic, intentional samples. The first was formed by 239 students (55.6% female, 44.4% male; age: 10.89 ± .66 years) from four Primary schools. The second was formed by 852 students (51.1% female, 48.9% male; age: 14.9 ± .1.75 years) from eight Secondary and Baccalaureate schools.

### Procedures

The questionnaire was administered collectively to students of primary, compulsory secondary and baccalaureate schools during regular school hours. After giving the appropriate instructions and obtaining informed consent (center and families), all students voluntarily filled out the requested information. The explanatory instructions were the same in all the classrooms and were given by the same interviewer in order to avoid a bias factor. The questionnaires were voluntarily filled by the participants, who answered anonymously all the items. Prior informed consent of the participants, families and educational centers were obtained before the study took part. All this was carried out according to the ethical standards of the Declaration of Helsinki 1961 (revised in Tokyo in 1989 and Edinburgh in 2000) and approved by the Ethics Committee of the University of Vigo.

### Instruments

To carry out this study, *ad hoc* sociodemographic and academic variables were used. In addition, the *Self-Report of Reasons for the Practice of Physical Exercise (*AMPEF) questionnaire was administered, adapted to Spanish by Capdevila et al [39] from the *Exercise Motivations Inventory-2* (EMI-2) by Markland & Ingledew [40]. This questionnaire includes 48 items that adopt a Likert-type response format with ten alternatives (from not at all true: 0, to totally true: 10). In this study, it presents good reliability in the preadolescent period (Cronbach's Alpha = .91) and in adolescence (Cronbach's alpha = .96). Self-Report Questionnaire of Barriers to Physical Activity Practice (ABPEF) [39], adapted by Niñerola et al. [41]. This version consists of 17 items that are answered on a Likert scale ranging from 0 (reason unlikely to prevent me from performing physical activity in the next few weeks) to 10 (reason most likely to prevent me from performing physical activity) points. The psychometric properties of both scales have been analyzed in the Spanish context [42, 43], and verified through an exploratory and confirmatory factor analysis of the validity and reliability of the instrument. The AMPEF has a total of 48 items, distributed in eight factors: as factor 1: Prevention and positive health (P/PH) with

factorial weights between .52 and .71; factor 2: Competition/ Social Recognition/ Challenge (C/SR/C) with factorial weights between .43 and .72; factor 3: Weight and Body Image (W/BI) with factorial weights ranging from .65 to .74; factor 5: Stress Control (SC) with factorial weights between .63 and .70; factor 6: Muscular Strength and Endurance (S/E) with factorial weights ranging between .68 and .73; factor 7: Health Emergencies (HE) with factorial weights ranging between .53 and .56; an factor 8: Agility and Flexibility (A/F) with factorial weights between .59 and .78 [42].

The ABPEF has four factors: Factor 1: Body Image/Physicalsocial Anxiety (BI/PSA) (with values of factorial loadings within a range of .634 to .864; Factor 2: Fatigue/Laziness (F/L) (with values of factorial loadings within a range from .422 to .767; Factor 3: Mandatories/Lack of Time (M/LT) (with values of factorial loadings within a range from .637 and .866; and Factor 4: Environment/Infrastructures (E/I) (with values of factorial loadings within a range from .421 and .712 [43].

## Data analysis

All the analyses were performed with the statistical package SPSS v.23 (Chicago, IL, USA). Firstly, the mean scores, standard deviation and t-test for unpaired samples were calculated to verify significant differences, when $p \leq .05$. Pearson's correlations were performed to assess the linear association between the variables. Cohen's $d$ was calculated to add accuracy to the data obtained, considering the effect size (small: .2; medium: .5; large: .8) [44].

## Results

Initially, in Table 1, the participants' mean scores on the motivational scale are shown according to their age. A coincidence is observed with regard to competition, social recognition and challenge as the most highly rated motivational factor. However, they differ in the order of the second and third factors with the highest average, which are prevention / positive health and affiliation / fun / well-being in preadolescents and affiliation / fun / well-being and prevention / positive health for adolescents. Likewise, there are also coincidences in the three factors with the least motivational importance (stress control, health emergencies, agility and flexibility), with few differences in the order of preference.

The comparison between the mean scores of the two groups of students showed statistically significant differences between the groups in the case of the following factors: prevention and positive health ($t = 5.24$, $p < .001$); competition, social recognition and challenge ($t = -16.02$, $p < .001$); affiliation, fun and well-being ($t = -19.54$, $p < .001$); stress control ($t = 5.57$, $p < .001$);

**Table 1. Descriptive analysis and differences between preadolescents and adolescents in motivation towards the practice of physical activity.**

| MOTIVACTIONAL FACTORS | Total (n = 1081) | | Preadolescence (n = 239) | | Adolescence (n = 842) | | t | p | d |
|---|---|---|---|---|---|---|---|---|---|
| | M | SD | M | SD | M | SD | | | |
| Prevention and positive health (P/PH) | 34.61 | 10.49 | 32.73 | 5.53 | 33.71 | 10.56 | 5.24 | .000 | -.116 |
| Competition, social recognition and challenge (C/SR/C) | 39.02 | 18.71 | 38.23 | 9.69 | 44.46 | 16.80 | -16.02 | .000 | -.454 |
| Body weight and body image (BW/BI) | 26.92 | 11.30 | 18.25 | 7.14 | 26.59 | 12.35 | -.25 | .803 | -.826 |
| Affiliation, amusement and well-being (A/A/WB) | 29.19 | 15.98 | 28.07 | 5.35 | 34.71 | 12.94 | -19.54 | .000 | -.671 |
| Stress control (SC) | 11.67 | 5.43 | 7.69 | 3.65 | 11.07 | 5.57 | 4.89 | .000 | -.718 |
| Strength and muscular endurance (S/ME) | 15.12 | 7.50 | 14.12 | 4.11 | 16.91 | 7,17 | -13.14 | .000 | -.477 |
| Health emergencies (HE) | 10.41 | 5.89 | 8.29 | 3.84 | 10.29 | 6.39 | 2.79 | .005 | -.759 |
| Agility and flexibility (A/F) | 11.15 | 5.24 | 10.89 | 3.25 | 11.98 | 5.31 | -6.95 | .000 | -.248 |

**Table 2. Descriptive analysis and differences between preadolescents and adolescents in barriers towards the practice of physical exercise.**

| BARRIERS | Total (n = 1081) | | Preadolescence (n = 239) | | Adolescence (n = 842) | | t | p | d |
|---|---|---|---|---|---|---|---|---|---|
| | M | SD | M | SD | M | SD | | | |
| Body image or physical social anxiety (BI/PSA) | 10.91 | 8.73 | 8.03 | 3.53 | 11.72 | 9.55 | -5.71 | .000 | -.513 |
| Fatigue or laziness (F/L) | 14.43 | 9.07 | 8.59 | 3.50 | 15.07 | 9.47 | -13.20 | .000 | -.908 |
| Mandatories or lack of time (M/LT) | 9.86 | 6.53 | 5.72 | 2.34 | 10.31 | 6.61 | -14.20 | .000 | -.925 |
| Environment or infrastructures (E/I) | 3.49 | 2.51 | 2.75 | 1.37 | 3.69 | 2.70 | -5.90 | .000 | -.439 |

muscle strength and endurance ($t$ = -13.14, $p <$ .001); health emergencies ($t$ = 2.79, $p <$ .05); and agility or flexibility ($t$ = -6.95, $p <$ .001). Thus, adolescents showed a significantly higher motivation than preadolescents in the practice of physical exercise. The effect size of the difference was considered very low in P/PH ($d$ = -.116) and A/F ($d$ = -.248), moderate in C/SR/C ($d$ = -.454), F/RM ($d$ = -.447), A/D/B ($d$ = -.671) and high in CE ($d$ = -.718), US ($d$ = -.759) and P/PH ($d$ = -.826). Finally, the motivational factor weight and body image did not produce significant differences between the averages ($t$ = -.25, $p >$.05).

In the case of barriers that hinder the practice of physical exercise (Table 2), fatigue or laziness is confirmed as the main barrier and the environment or facilities are less influential for both preadolescents and adolescents. In this scale, significant differences are obtained for all the factors [body image or physical-social anxiety (t = -5.71, $p <$ .001), fatigue or laziness (t = -13.20, $p <$ .001), obligations or lack of time (t = -14.20, $p <$ .001) and environment or facilities (t = -5.90, $p <$ .001)]. As a result, there is a greater presence of barriers to the practice of physical exercise in the adolescent period. The effect size was moderate in A / I ($d$ = -. 439) and IC / AFS ($d$ = -. 513) and very high in F / P ($d$ = -. 908) and O / FT ($d$ = -.925).

The study of correlation between the motivational scale scores (Table 3) showed a better correlation of the factors in the adolescent period ($r$ between .307 and .703) than in the preadolescent period ($r$ between .172 and .523). Note that the highest correlation, both in preadolescents ($r$ = .523) and adolescents ($r$ = .703) occurs between competition, social recognition or challenge and strength or muscular endurance. There is an absence of correlation in the preadolescent period between the factors weight or body image and affiliation, fun or well-being ($r$ = .052, $p >$.05) and between affiliation, fun or well-being and health emergencies ($r$ = -.004, $p >$.05).

**Table 3. Correlations for preadolescents and adolescents in motivation towards the practice of physical exercise.**

| MOTIVATIÓN | P/PH | C/SR/C | BW/BI | A/A/BW | SC | S/ME | HE | A/F |
|---|---|---|---|---|---|---|---|---|
| P/PH | | .441** | .422** | .318** | .300** | .448** | .229** | .377** |
| C/SR/C | .573** | - | .415** | .406** | .292** | .523** | .236** | .389** |
| BW/BI | .507** | .443** | - | .052 | .424** | .473** | .464** | .384** |
| A/A/BW | .678** | .743** | .307** | - | .266** | .248** | -.004 | .172** |
| SC | .625** | .520** | .435** | .509** | - | .250** | .470** | .231** |
| S/ME | .681** | .703** | .465** | .663** | .500** | - | .198** | .391** |
| HE | .494** | .509** | .410** | .412** | .466** | .417** | - | .183** |
| A/F | .596** | .584** | .488** | .529** | .429** | .622** | .443** | - |

Note: The correlations above the diagonal belong to the sample of children (n = 239), while those below the diagonal belong to the sample of adolescents (n = 842).

Significant correlation to levels

** p < .01

**Table 4. Correlations for preadolescents and adolescents in barriers to the practice of physical exercise.**

| *BARRIERS* | IC/AFS | F/P | O/FT | A/I |
|---|---|---|---|---|
| IC/AFS | - | .632** | .441** | .377** |
| F/P | .531** | - | .521** | .360** |
| O/FT | .206** | .292** | - | .549** |
| A/I | .409** | .346** | .241** | - |

Note: The correlations above the diagonal belong to the sample of preadolescents (N = 239) and those below the diagonal belong to the sample of adolescents (N = 842). Significant correlation to levels

** $p < .01$

Finally, the scale for barriers to the practice of physical exercise (Table 4) shows a slightly higher correlation in the preadolescent period (r between .360 and .632) than in the adolescent period (r between .206 and .531). The greatest correlation is observed between fatigue or laziness and body image or physical-social anxiety, both in the preadolescents (r = .632) and in the adolescents (r = .531).

## Discussion and conclusions

Despite the diverse possibilities and variables for study in the field of physical exercise, the aim of this study was to examine the reasons for and barriers to the practice of physical exercise in preadolescents and adolescents, verifying the existence of significant differences between these two periods. When we speak about physical exercise as a predictor of quality of life with health and educational implications, we consider that it could help reduce sedentarism in children and young people [45].

The results obtained partially confirm our first hypothesis, since they present a higher motivation and the presence of barriers in the adolescent period. Likewise, the data indicate that competition, social recognition and challenge are the most important motivational aspects for the preadolescent and adolescent period, coinciding with the literature reviewed [46]. According to these authors, the motivational attitude towards physical activity is mainly focused on perceptions of competence and body image [47], fun [48], influence of friends [49], relatives [25] or Physical Education teachers [50] and opportunities for physical activity in the natural environment [51]. However, these results differ from other studies [11, 52] that indicate enjoyment, fun and occupation of free time as the main motivations. This may be due to work done in different countries or age groups, since the reasons change as age progresses. In contrast, health emergencies and agility or flexibility are specified as the motivational reasons for less frequency of exercise.

Following the theory of Self-determination [52], people are whether involved or not in physical activity based on a series of psychological mechanisms that regulate this behaviour, and seeking as far as possible for a greater orientation towards the self-determined motivation. Studies show that the theory of Self-determination motivation is highly related to intrinsic motivation, while extrinsic motivation and its regulatory mechanisms favour behaviours that are not self-determined and are even characterized by a lack of motivation [53, 54].

Taking into account the barriers, the results show fatigue or laziness as the most recurrent factor, confirming its increase with age (greater presence in adolescence). These findings are in line with previous investigations that indicate fatigue or laziness, lack of time, work or family demands and lack of money as the main barriers to physical exercise [55, 56]. On the other hand, the environment or facilities are the least important barriers for preadolescents and adolescents. Studies such as the one conducted by Kwan & Faulkner [56] in university populations

show a perception of more barriers at the start of the degree program (younger age) than in the final years (older).

With regard to the inferential analysis carried out, the motivational factors were significantly differentiated according to age, especially when referring to affiliation, fun, well-being, competition, social recognition and challenge. Thus, adolescents were more motivated than preadolescents, which implies greater interest and cognitive involvement in the practice of physical exercise as age advances. Likewise, barriers to physical exercise also showed significant differences between the two periods, with greater occurrence of barriers in adolescence, especially with respect to fatigue, laziness, obligations and lack of time. There are numerous studies that have shown a variation in physical exercise with age, mainly noting a decline in exercise as people get older [57, 58].

On the other hand, the second hypothesis is not confirmed because although there is a relationship between motivational factors and barriers, these relationships are weak. Thus, correlational analysis has found that motivational factors are better related in the adolescent period; whereas, in the case of barriers, the opposite occurs and the relationship is more intense in the preadolescent stage. However, all the factors (motivational and barriers) showed small yet positive and significant relationships. From this, it is inferred that an increase or decrease in one motivational factor or barrier helps to improve or worsen the rest. Along the same lines, Rodríguez-Romo et al. [59] report that the reasons that lead individuals to practice or abandon exercise are variable and are difficult to attribute to a single reason.

One of the limitations is that motivation or barriers to the practice of physical exercise are not observable variables, but "a hypothetical construct that starts from inferences made from behavioral manifestations" [60]. In fact, establishing a more robust relationship between motivational factors or barriers and other sociodemographic variables, broadening the age range of the participants and taking into consideration other agents within the context (teachers and families) are aspects that should be considered for future studies. Therefore, considering the importance of physical exercise at young ages, these findings will allow us to design specific actions for each period taking into account the specific characteristics of the population sector and to achieve greater adherence to physical exercise programs.

## Implications for school health

The results of our research suggest the need to control barriers to the practice of physical exercise, promoting the motivational aspects that most appeal to adolescent behaviors, mainly in the preadolescent period. Interventions to increase physical activity must start from adolescence to consolidate healthy habits. Also, students must perceive the incentive to use their leisure time to do physical exercise [54]. The possibility of creating a new professional profile that already exists in other countries, namely kinesiology, should be assessed. A professional kinesiologist could be a mediator during school leisure time, as well as in other community dynamics, and even at health centers for other populations.

## Author Contributions

**Conceptualization:** Iago Portela-Pino, María José Martínez-Patiño.

**Data curation:** Iago Portela-Pino, Teresa Valverde-Esteve.

**Formal analysis:** Iago Portela-Pino, Teresa Valverde-Esteve.

**Investigation:** Iago Portela-Pino, Teresa Valverde-Esteve, María José Martínez-Patiño.

**Methodology:** Iago Portela-Pino, Teresa Valverde-Esteve.

**Project administration:** María José Martínez-Patiño.

**Resources:** Iago Portela-Pino, María José Martínez-Patiño.

**Supervision:** Teresa Valverde-Esteve, María José Martínez-Patiño.

**Validation:** María José Martínez-Patiño.

**Visualization:** María José Martínez-Patiño.

**Writing – original draft:** Iago Portela-Pino, Teresa Valverde-Esteve, María José Martínez-Patiño.

**Writing – review & editing:** Iago Portela-Pino, Teresa Valverde-Esteve, María José Martínez-Patiño.

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
