## [Decision Letter · Decision Letter 0]

30 Mar 2021

PONE-D-21-07811

AGE AS A DETERMINING VARIABLE IN THE PRACTICE OR ABANDONMENT OF PHYSICAL EXERCISE AMONG YOUNG PEOPLE

PLOS ONE

Dear Dr. Valverde,

Thank you for submitting your manuscript to PLOS ONE. After careful consideration, we feel that it has merit but does not fully meet PLOS ONE’s publication criteria as it currently stands. Therefore, we invite you to submit a revised version of the manuscript that addresses the points raised during the review process.

We look forward to receiving your revised manuscript.

Kind regards,

José Gutiérrez-Pérez

Academic Editor

PLOS ONE

Journal Requirements:

Could you therefore please include the title page into the beginning of your manuscript file itself, listing all authors and affiliations

3. Thank you for including your ethics statement:  "Ethics Committee of the University of Vigo".   

Please provide additional details regarding participant consent. In the ethics statement in the Methods and online submission information, please ensure that you have specified what type you obtained (for instance, written or verbal, and if verbal, how it was documented and witnessed). If your study included minors, state whether you obtained consent from parents or guardians. If the need for consent was waived by the ethics committee, please include this information.

4. Please include your tables as part of your main manuscript and remove the individual files. Please note that supplementary tables (should remain/ be uploaded) as separate "supporting information" files

Additional Editor Comments (if provided):

The paper analyses the fact that there is a decline in the practice of Physical Exercise during adolescence, lack of time is the main barrier in the practice of physical exercise; and adds that competition, social recognition and challenge are the most important motivational aspects for the preadolescent and adolescent period; fatigue or laziness as the most recurrent factor as a barrier, confirming its increase with age.

The results suggest the need to control barriers to the practice of physical exercise, promoting the interventions on motivational aspects that most appeal to adolescent behaviors, mainly in the preadolescent period to increase physical activity and consolidate healthy habits.

It would be desirable to include more details of the factorial scores in relation to the different factors evaluated and how the variables saturate them. A table that includes variance percentages and eigenvalues for each factor would provide the paper with higher quality.

The manuscript requires major changes, as suggested by the two reviews.

Reviewers' comments:

Reviewer's Responses to Questions

**Comments to the Author**

1. Is the manuscript technically sound, and do the data support the conclusions?

Reviewer #1: Partly

Reviewer #2: Partly

2. Has the statistical analysis been performed appropriately and rigorously? 

Reviewer #1: Yes

Reviewer #2: Yes

3. Have the authors made all data underlying the findings in their manuscript fully available?

Reviewer #1: Yes

Reviewer #2: Yes

4. Is the manuscript presented in an intelligible fashion and written in standard English?

Reviewer #1: Yes

Reviewer #2: Yes

5. Review Comments to the Author

Reviewer #1: The article has strengths that endow it with quality, but below are a series of comments that I consider should be taken into account to provide more quality and coherence to the manuscript:

It is recommended to incorporate in the introduction and in the justification of the problem, a more up-to-date discourse highlighting the importance of this study in the current context, even making reference to the work that has to be done from education in favor of quality education and within the framework of compliance with the 2030 Agenda and the SDGs; influencing those SDGs that are closely related to the issue addressed.

In the methodological framework, it is recommended that the authors explain and justify in a broader way the methodological approach of the research, as well as the paradigm from which the study is contextualized.

Explain in more detail the proposed sociodemographic variables as well as a more detailed explanation of the instrument used, taking into account the dimensions and themes in which the items are structured.

It is not clear if the factors used by the authors are due to the fact that the questionnaire is structured in this way or because they have carried out factor analyzes. In which case we recommend explaining it better and, in that case, doing ANOVA. Therefore, better justify the analysis procedure to better understand the results obtained.

It is recommended to include a bibliographic citation in the statements that are detailed in the section “Implications for school health”.

Reviewer #2: General comments:

The practice of physical activity is deeply related to quality of life and satisfaction with it, being particularly important in the field of education and health. Consequently, in order to achieve a better quality of life, it is essential to know about changes in behavior, action planning, identification of barriers, resolution of difficulties or motivational aspects that lead children and adolescents to increase the practice of physical activity. These ages are a key period for the acquisition of habits and routines. However, the current article raises some questions due to the sample, the instrument, the lack of critical reflection, current review of the literature in the educational field, etc. Despite the interesting topic and approach, the article at this point should review some main elements.

Some main comments:

Introduction: The introduction would benefit from information on the theory of self-determination and also from paragraphs showing the implications of the educational setting for the subject of study.

Method: There is a lot of difference in the participation of schools (n = 239) and of secondary and high school (n = 852) in the study, which could determine the results found. It would be important to explain why. As for the instrument, its version used (Capdevilla, Niñerola and Pintanel, 2004) was carried out with adolescents and adults, but in this study it has been used with children and adolescents, therefore, for the Primary I do not know if it would be the most suitable instrument. In addition, it is not indicated in the study procedure if some type of adaptation has been made for completion in Primary schools.

Discussion: The discussion and the “implications for school health” could, for example, focus on the theory of self-determination to justify some of the results found and possible future recommendations. In addition, the article would also benefit from commenting on effective strategies that Primary and Secondary schools have carried out in recent years to improve study variables, with the involvement of teachers, families, peer teaching/mentoring, comprehensive programs and collaborative creating bridges for example between universities, schools, sports clubs, etc. It is considered the implications of starting physical activity interventions in adolescence, right before those ages? This idea is not understood based on the results found in the study and the existing literature.

References: Introduce more current references in some cases.

6. PLOS authors have the option to publish the peer review history of their article (what does this mean?). If published, this will include your full peer review and any attached files.

Reviewer #1: No

Reviewer #2: No

---

## [Author Response · Author response to Decision Letter 0]

6 Apr 2021

Dear Editor, 

The authors would like to thank you for providing us the opportunity to revise and improve our manuscript with the suggestions provided by the referees. 

Here is our response to your comments, 

Best wishes, 

Journal Requirements:

1. Please ensure that your manuscript meets PLOS ONE's style requirements, including those for file naming. The PLOS ONE style templates can be found at https://journals.plos.org/plosone/s/file?id=wjVg/PLOSOne_formatting_sample_main_body.pdf and https://journals.plos.org/plosone/s/file?id=ba62/PLOSOne_formatting_sample_title_authors_affiliations.pdf Thank you, amended. 

Could you therefore please include the title page into the beginning of your manuscript file itself, listing all authors and affiliations Thank you, amended. Sorry for the inconvenience. 

3. Thank you for including your ethics statement: "Ethics Committee of the University of Vigo". 

Please provide additional details regarding participant consent. In the ethics statement in the Methods and online submission information, please ensure that you have specified what type you obtained (for instance, written or verbal, and if verbal, how it was documented and witnessed). If your study included minors, state whether you obtained consent from parents or guardians. If the need for consent was waived by the ethics committee, please include this information. Thank you. We have added the following statement: “The questionnaires were voluntarily filled by the participants, who answered anonymously all the items. Prior informed consent of the participants, families and educational centers were obtained before the study took part”.

Once you have amended this/these statement(s) in the Methods section of the manuscript, please add the same text to the “Ethics Statement” field of the submission form (via “Edit Submission”). Thank you. 

4. Please include your tables as part of your main manuscript and remove the individual files. Please note that supplementary tables (should remain/ be uploaded) as separate "supporting information" files. Amended, thank you. 

Additional Editor Comments (if provided):

The paper analyses the fact that there is a decline in the practice of Physical Exercise during adolescence, lack of time is the main barrier in the practice of physical exercise; and adds that competition, social recognition and challenge are the most important motivational aspects for the preadolescent and adolescent period; fatigue or laziness as the most recurrent factor as a barrier, confirming its increase with age.

The results suggest the need to control barriers to the practice of physical exercise, promoting the interventions on motivational aspects that most appeal to adolescent behaviors, mainly in the preadolescent period to increase physical activity and consolidate healthy habits.

It would be desirable to include more details of the factorial scores in relation to the different factors evaluated and how the variables saturate them. A table that includes variance percentages and eigenvalues for each factor would provide the paper with higher quality. varia

The manuscript requires major changes, as suggested by the two reviews.

Reviewers' comments:

Reviewer's Responses to Questions

Comments to the Author

1. Is the manuscript technically sound, and do the data support the conclusions?

Reviewer #1: Partly

Reviewer #2: Partly

2. Has the statistical analysis been performed appropriately and rigorously? 

Reviewer #1: Yes

Reviewer #2: Yes

3. Have the authors made all data underlying the findings in their manuscript fully available?

Reviewer #1: Yes

Reviewer #2: Yes

4. Is the manuscript presented in an intelligible fashion and written in standard English?

Reviewer #1: Yes

Reviewer #2: Yes

5. Review Comments to the Author

Reviewer #1: The article has strengths that endow it with quality, but below are a series of comments that I consider should be taken into account to provide more quality and coherence to the manuscript:

It is recommended to incorporate in the introduction and in the justification of the problem, a more up-to-date discourse highlighting the importance of this study in the current context, even making reference to the work that has to be done from education in favor of quality education and within the framework of compliance with the 2030 Agenda and the SDGs; influencing those SDGs that are closely related to the issue addressed.

In the methodological framework, it is recommended that the authors explain and justify in a broader way the methodological approach of the research, as well as the paradigm from which the study is contextualized. Amended, thanks. 

Explain in more detail the proposed sociodemographic variables as well as a more detailed explanation of the instrument used, taking into account the dimensions and themes in which the items are structured.

It is not clear if the factors used by the authors are due to the fact that the questionnaire is structured in this way or because they have carried out factor analyzes. In which case we recommend explaining it better and, in that case, doing ANOVA. Therefore, better justify the analysis procedure to better understand the results obtained.

It is recommended to include a bibliographic citation in the statements that are detailed in the section “Implications for school health”.

Reviewer #2: General comments:

The practice of physical activity is deeply related to quality of life and satisfaction with it, being particularly important in the field of education and health. Consequently, in order to achieve a better quality of life, it is essential to know about changes in behavior, action planning, identification of barriers, resolution of difficulties or motivational aspects that lead children and adolescents to increase the practice of physical activity. These ages are a key period for the acquisition of habits and routines. However, the current article raises some questions due to the sample, the instrument, the lack of critical reflection, current review of the literature in the educational field, etc. Despite the interesting topic and approach, the article at this point should review some main elements.

Some main comments:

Introduction: The introduction would benefit from information on the theory of self-determination and also from paragraphs showing the implications of the educational setting for the subject of study.

Method: There is a lot of difference in the participation of schools (n = 239) and of secondary and high school (n = 852) in the study, which could determine the results found. It would be important to explain why. As for the instrument, its version used (Capdevilla, Niñerola and Pintanel, 2004) was carried out with adolescents and adults, but in this study it has been used with children and adolescents, therefore, for the Primary I do not know if it would be the most suitable instrument. In addition, it is not indicated in the study procedure if some type of adaptation has been made for completion in Primary schools.

Discussion: The discussion and the “implications for school health” could, for example, focus on the theory of self-determination to justify some of the results found and possible future recommendations. In addition, the article would also benefit from commenting on effective strategies that Primary and Secondary schools have carried out in recent years to improve study variables, with the involvement of teachers, families, peer teaching/mentoring, comprehensive programs and collaborative creating bridges for example between universities, schools, sports clubs, etc. It is considered the implications of starting physical activity interventions in adolescence, right before those ages? This idea is not understood based on the results found in the study and the existing literature.

References: Introduce more current references in some cases.

---

## [Decision Letter · Decision Letter 1]

18 May 2021

PONE-D-21-07811R1

AGE AS A DETERMINING VARIABLE IN THE PRACTICE OR ABANDONMENT OF PHYSICAL EXERCISE AMONG YOUNG PEOPLE

PLOS ONE

Dear Dr. Valverde,

Thank you for submitting your review manuscript to PLOS ONE. After careful consideration, we feel that it has merit but does not fully meet PLOS ONE’s publication criteria as it currently stands. Therefore, we invite you to submit a second revised version of the manuscript that addresses the points raised during the review process.

After a second round of review, there is some improvement effort, but essential aspects regarding the objections raised by the reviewers have not been addressed. You must make an exhaustive effort to address all the improvements made, the paper requires greater dedication.

We look forward to receiving your revised manuscript.

Kind regards,

José Gutiérrez-Pérez

Academic Editor

PLOS ONE

Reviewer's Responses to Questions

**Comments to the Author**

Reviewer #1: After reviewing the new version of the manuscript provided by the authors, I consider that they have adequately addressed the theoretical changes but not the methodological changes.

They justify with bibliographic citations the criteria of reliability and validity of the instrument used, although it is missing that they expose the factors from which they start taking into account the previous validation studies. On the other hand, they do not carry out more complex statistical analyzes from which the fulfillment of the objective of the study is really seen. The authors carry out basic descriptive analyzes and the correlational analyzes they propose are weak, so it is recommended to carry out another type of analysis that gives strength and quality to the results obtained.

Reviewer #2: The introduction and discussion would benefit from information on the theory of self-determination, but there is no change, proposal or reference in relation to this comment. Although the authors have answered that it had been amended in the new version.

---

## [Author Response · Author response to Decision Letter 1]

19 May 2021

Dear editor, 

The authors would like to thank you and the reviewers for providing us such constructive feedback and the opportunity. Here is our response. 

Yours faithfully, 

Reviewer's Responses to Questions

Comments to the Author

Reviewer #1: After reviewing the new version of the manuscript provided by the authors, I consider that they have adequately addressed the theoretical changes but not the methodological changes.

They justify with bibliographic citations the criteria of reliability and validity of the instrument used, although it is missing that they expose the factors from which they start taking into account the previous validation studies. Thanks for your comment. The following aspects have been described in the text: 

The AMPEF has a total of 48 items, distributed in eight factors: as factor 1: Prevention and positive health (P/PH) with factorial weights between .52 and .71; factor 2: Competition/ Social Recognition/ Challenge (C/SR/C) with factorial weights between .43 and .72; factor 3: Weight and Body Image (W/BI) with factorial weights ranging from .65 to .74; factor 5: Stress Control (SC) with factorial weights between .63 and .70; factor 6: Muscular Strength and Endurance (S/E) with factorial weights ranging between .68 and .73; factor 7: Health Emergencies (HE) with factorial weights ranging between .53 and .56; an factor 8: Agility and Flexibility (A/F) with factorial weights between .59 and .78. 

The ABPEF has four factors: Factor 1: Body Image/Physicalsocial Anxiety (BI/PSA) (with values of factorial loadings within a range of .634 to .864; Factor 2: Fatigue/Laziness (F/L) (with values of factorial loadings within a range from .422 to .767; Factor 3: Mandatories/Lack of Time (M/LT) (with values of factorial loadings within a range from .637 and .866; and Factor 4: Environment/Infrastructures (E/I) (with values of factorial loadings within a range from .421 and .712.

On the other hand, they do not carry out more complex statistical analyzes from which the fulfillment of the objective of the study is really seen. The authors carry out basic descriptive analyzes and the correlational analyzes they propose are weak, so it is recommended to carry out another type of analysis that gives strength and quality to the results obtained.

Thanks for the comment. The aim of this study was to investigate the motivations or barriers that lead to greater or lesser adherence to physical exercise in children and adolescents, according to the age. Therefore, the authors consider that the analysis respond to the objectives and hypothesis raised. 

Reviewer #2: The introduction and discussion would benefit from information on the theory of self-determination, but there is no change, proposal or reference in relation to this comment. Although the authors have answered that it had been amended in the new version.

Thanks for your comment. We have added the following paragraph in the manuscript:

Following the theory of Self-determination (Deci and Ryan, 2002), people are whether involved or not in physical activity based on a series of psychological mechanisms that regulate this behaviour, and seeking as far as possible for a greater orientation towards the self-determined motivation. Studies show that the theory of Self-determination motivation is highly related to intrinsic motivation, while extrinsic motivation and its regulatory mechanisms favour behaviours that are not self-determined and are even characterized by a lack of motivation (Moreno & Martínez, 2006; Salazar-Ayala, & Gastélum-Cuadras, 2020).

---

## [Editor Report · Decision Letter 2]

14 Jun 2021

AGE AS A DETERMINING VARIABLE IN THE PRACTICE OR ABANDONMENT OF PHYSICAL EXERCISE AMONG YOUNG PEOPLE

PONE-D-21-07811R2

Dear Dr. Valverde,

We’re pleased to inform you that your manuscript has been judged scientifically suitable for publication and will be formally accepted for publication once it meets all outstanding technical requirements.

Kind regards,

José Gutiérrez-Pérez

Academic Editor

PLOS ONE

---

## [Editor Report · Acceptance letter]

16 Jun 2021

PONE-D-21-07811R2 

Age as a Determining Variable in the Practice or Abandonment of Physical Exercise among Young People 

Dear Dr. Valverde-Esteve:

I'm pleased to inform you that your manuscript has been deemed suitable for publication in PLOS ONE. Congratulations! Your manuscript is now with our production department. 

Kind regards, 

on behalf of

Dr. José Gutiérrez-Pérez 

Academic Editor

PLOS ONE